# Pharmacokinetic Characterization of (Poly)phenolic Metabolites in Human Plasma and Urine after Acute and Short-Term Daily Consumption of Mango Pulp

**DOI:** 10.3390/molecules25235522

**Published:** 2020-11-25

**Authors:** Jiayi Fan, Di Xiao, Liyun Zhang, Indika Edirisinghe, Britt Burton-Freeman, Amandeep K. Sandhu

**Affiliations:** Department of Food Science and Nutrition, Center for Nutrition Research, Institute for Food Safety and Health, Illinois Institute of Technology, Chicago, IL 60616, USA; fjiayi@hawk.iit.edu (J.F.); dxiao6@iit.edu (D.X.); lzhan134@hawk.iit.edu (L.Z.); iedirisi@iit.edu (I.E.); bburton@iit.edu (B.B.-F.)

**Keywords:** mango, polyphenols, gallotannins, pharmacokinetics, plasma, urine

## Abstract

Pharmacokinetic (PK) evaluation of polyphenolic metabolites over 24 h was conducted in human subjects (*n* = 13, BMI = 22.7 ± 0.4 kg/m^2^) after acute mango pulp (MP), vitamin C (VC) or MP + VC test beverage intake and after 14 days of MP beverage intake. Plasma and urine samples were collected at different time intervals and analyzed using targeted and non-targeted mass spectrometry. The maximum concentrations (C_max_) of gallotannin metabolites were significantly increased (*p* < 0.05) after acute MP beverage intake compared to VC beverage alone. MP + VC beverage non-significantly enhanced the C_max_ of gallic acid metabolites compared to MP beverage alone. Pyrogallol (microbial-derived metabolite) derivatives increased (3.6%) after the 14 days of MP beverage intake compared to 24 h acute MP beverage intake (*p* < 0.05). These results indicate extensive absorption and breakdown of gallotannins to galloyl and other (poly)phenolic metabolites after MP consumption, suggesting modulation and/or acclimation of gut microbiota to daily MP intake.

## 1. Introduction

Mango (*Mangifera indica* L.) is one of the most economically important fruits in the world due to its appealing taste and high nutritional value [1]. Mango has a unique and diverse phytochemical profile [2] consisting of carotenoids and a number of polyphenol compounds including mangiferin, gallotannins, catechin, quercetin, kaempferol, gallic acid, ellagic acid and other phenolic acids [3]. The polyphenolic profile of mango is associated with reducing the risk of developing a number of chronic diseases and their related complications [4,5]. Several in vitro and in vivo animal studies support anti-diabetic [6,7], anti-cancer [8,9], anti-inflammatory [10,11], anti-oxidant [12,13] and anti-bacterial [14] activities linked with the intake of mango pulp, peel, seed, juice, extracts and other mango products. Additionally, bioavailability and bioaccessibility analyses of isolated compounds (mangiferin) or extracts from mango leaf and mango seed kernel have been conducted in pre-clinical and in vitro models [15,16,17,18,19]. However, mango pulp (MP) consumed by humans involves delivery/absorption of bioactive components from a complex food matrix potentially influencing bioavailability. Only a few studies have assessed the absorption and metabolism of polyphenols from mango fruit in humans [4,20,21].

A previous study identified and quantified five compounds in plasma and seven compounds in urine after “Ataulfo” mango consumption (500 g of flesh or 721 g of juice) [21]. Barnes et al. reported seven galloyl metabolites in urine after a 10-day consumption of 400 g of MP from the Keitt variety [20]. The same research group reported five galloyl-derivatives in plasma after 6 weeks of mango supplementation [4]. Collectively, only a few metabolites have been reported in plasma after MP intake and mangiferin has never been detected in previous studies.

Therefore, the first objective of this study was to characterize and investigate the absorption/kinetic profile of mango (poly)phenols and their metabolites in plasma over 24 h after MP consumption.

Black pepper (piperine), vitamin C (VC) and vitamin E have all been documented to enhance the absorption of polyphenol compounds when added to foods and beverages [22,23]. Vitamin C enhanced the absorption of green tea polyphenols in humans [24] and worked synergistically with oat (poly)phenolics and almond skin flavonoids to protect low density lipoprotein (LDL) against oxidation in a hamster and human model [25,26]. Vitamin C is inherent to the mango fruit; however, amounts may not be sufficient to stabilize polyphenols during digestion and transit through the gastrointestinal tract to enhance absorption in the small intestine or help maintain structural integrity for microbial metabolism in the large intestine. So, the second objective of this study was to test the hypothesis that concomitant intake of VC with MP would enhance absorption of mango (poly)phenols by increasing the number and/or concentration of mango (poly)phenolic metabolites.

Our previous work with berries indicates that regular intake or exposure to fruit polyphenols increases concentration of some metabolites in blood [27,28,29,30]. Likewise, increased excretion of certain mango metabolites was observed in urine after a 10-day intake of MP, of which some were microbial-derived [20]. Therefore, the third objective of this study was to assess the influence of daily intake of MP for 14 days on (poly)phenolic metabolite profiles.

## 2. Results and Discussion

### 2.1. Mango Pulp

Among the 46 tentatively identified compounds in MP (Appendix A, Table A1), gallotannins accounted for a large proportion of total MP polyphenols, including galloyl diglucoside, galloyl glucose isomers, galloyl-quinic acid, trigalloyl glucose isomers, tetragalloyl glucose isomers, and pentagalloyl glucose. Besides gallic acid derivatives, mangiferin, the signature compound of mango [31], was also identified along with quercetin, kaempferol, benzoic acids, protocatechuic acids, catechin, coumarins and ellagic acid compounds. These compounds have also been reported previously in mangoes as well as in barks, kernels, peels and leaves [1,2,3,31,32].

The potential health benefits of various mango polyphenols are well documented in previous studies. For example, gallotannins and generated metabolites from intestinal microbial catabolism have been shown to possess anti-obesogenic effects in human and mouse models [4,33] and antibacterial effects in vitro [34,35]. Therefore, understanding the metabolic fate of mango polyphenols and their PK (pharmacokinetic) profiles after consumption will facilitate interpreting their roles in providing health benefits in humans.

### 2.2. Acute Study

The objective of the acute 24 h study was to characterize and determine PK parameters of mango polyphenols and the generated metabolites in human plasma after a single, one-time intake of a MP beverage. The acute study also aimed to evaluate the potential of VC to enhance absorption of MP polyphenols. The (poly)phenolic metabolites were identified using UHPLC-Q-TOF-MS in the pooled plasma (*n* = 13) at time points 0, 2, 8 and 24 h. A total of 18 (poly)phenolic metabolites were characterized in the pooled plasma samples at 2, 8 and 24 h after MP beverage intake (Table 1).

For the acute part of the study the quantitative analysis was conducted for 11 (poly)phenolic compounds and their metabolites in plasma. The mean plasma concentration vs. time profiles (0−24 h) are reported for 10 metabolites that were detected above the limit of quantitation (LOQ): gallic acid (**C1**), galloyl glucose (**C2**), methylgallic acid (**C3**), methylgallic acid sulfate (**C4**), ferulic acid hexoside (**C5**), methylpyrogallol sulfate isomer 1 (**C6**), methylpyrogallol sulfate isomer 2 (**C7**), pyrogallol sulfate (**C8**), methylpyrogallol glucuronide (**C9**) and catechol glucuronide (**C10**), after ingestion of MP, MP + VC and VC test beverages (Figure 1). The C_max_ and AUC_0–24h_ values of these 10 metabolites were significantly higher in plasma after MP and MP + VC beverages intake compared to VC beverage intake only except for methylpyrogallol glucuronide (**C9**) which failed the normality test for C_max_ during statistical analysis and had no significant differences in AUC_0–24h_ among treatments (Table 2). Mangiferin (**C11**) was detected but was below the LOQ. The pharmacokinetic curve of mangiferin is reported as the area vs. time profile (0−24 h) instead of mean plasma concentration vs. time profile (Figure 2). Mangiferin had significantly higher content in plasma (as observed from areas) after MP and MP + VC beverages intake compared to VC beverage intake alone (Table 2). The presence of gallic acid and mangiferin in plasma after MP beverage intake was confirmed by matching the MRM (multiple-reaction monitoring) ion transitions of the gallic acid and mangiferin standard with the compounds detected in the plasma. The identification of plasma metabolites including galloyl glucose, methylgallic acid, methylgallic acid sulfate, ferulic acid hexoside and methylpyrogallol glucuronide was confirmed by UHPLC-Q-TOF-MS and previous literature reports [2,20].

In general, (poly)phenolic compounds undergo structural modification before being absorbed into the blood. Compounds that escape absorption from the small intestine proceed to the colon where they are converted to various small molecules (phenolic acids, valerolactones, urolithins, etc.) by micro-organisms present in the lower bowel [36,37]. The metabolic products from the colon and the deconjugated (poly)phenols and aglycone structures from the upper digestive tract undergo phase II metabolism in the small intestine, liver and/or kidney resulting in methylated, glucuronidated and sulfoconjugated metabolites [38,39], while phase I metabolism (oxidation/reduction reactions) occurs to a lesser extent [40]. The resulting metabolites circulate in the blood and are transported to various body tissues and organs. Finally, the majority of metabolites are excreted in the urine by the kidneys [41].

Similar to mango pulp composition, most of the quantified compounds in plasma after acute supplementation of mango were gallic acid derivatives (Table 2, Figure 1 and Figure 2). Free gallic acid, galloyl glucose and gallic acid released from gallotannins were absorbed within 1–2 h and formed methyl, sulfate and glucuronide metabolites. Compounds such as gallic acid and its derivatives (**C1**–**C4**), ferulic acid hexoside (**C5**) and mangiferin (**C11**) reached their maximum concentration/area in plasma within 1–2 h after mango intake, and were cleared from the bloodstream within 6–8 h, suggesting absorption from the small intestine. On the other hand, several other metabolites (**C6**–**C10**) peaked at a much later time (8–10 h), suggesting microbial metabolism and absorption from the lower bowel. Pyrogallol is a major microbial metabolite of gallotannins [42] and gallic acid [43]. Several studies support a colonic origin of pyrogallol and its derivatives [21,33,44,45,46,47]. The conjugates of pyrogallol (**C6**–**C9**) achieved maximum concentrations in plasma between 6–10 h after mango intake. Catechol glucuronide (**C10**) showed a similar absorption pattern as pyrogallol derivatives. Inter-individual variability (high standard error) was observed in some metabolites especially those derived from gut microbial action possibly due to differences in the intestinal flora of individuals.

The enhancement of the absorption of mango polyphenols in the presence of VC was also explored in this acute study. There was an increase in the concentrations/areas of five metabolites, including gallic acid (**C1**), galloyl glucose (**C2**), methylgallic acid (**C3**), methylgallic acid sulfate (**C4**) and mangiferin (C11) after intake of MP + VC beverage compared to MP beverage alone (Figure 1 and Figure 2); however, the differences were not statistically significant (Table 2 and Figure 2). The polyphenols are prone to autoxidation when exposed to slightly basic environments such as the small intestine [48]. Vitamin C as an antioxidant may prevent the oxidation of mango polyphenols to some extent, such that simultaneous intake of mango with VC may reduce the gastrointestinal degradation of mango polyphenols. Previous studies have reported enhanced absorption of green tea polyphenols with VC. An increased absorption of epigallocatechin (EGC) and epigallocatechin gallate (EGCG) were observed in vitro [49], in vivo [50] and in humans [51] when green tea or its extracts were combined with VC-rich mixtures. In this study, even though higher concentrations/areas were observed for some metabolites in the MP + VC group compared to the MP group, the differences were not statistically significant. Further research with a higher dose of VC or combination with other bioactive compounds such as piperine [22] is needed to determine if dose or compound type could be the factors affecting absorption of (poly)phenolic compounds.

### 2.3. Short-Term 14 Day Mango Feeding Trial

In this part of the study, metabolite pool changes after daily intake of MP for 14 days were assessed. As noted in the acute study, several MP polyphenols are subjected to gut microbial metabolism. We hypothesized that with consistent daily intake/exposure to these polyphenols, an increase in generated metabolites would be observed due to an increase in microbial population or upregulation of the microbial mechanisms metabolizing the consistently available substrate.

A total of 166 (poly)phenolic metabolites were quantified in urine (Appendix A, Table A2). The compound which showed the highest urinary excretion when compared to fasting baseline urine was pyrogallol sulfate isomer (Δ = 1415.1 ± 955.2 nmol/L for 24 h fasting sample, Δ = 5298.0 ± 56.6 nmol/L for 15th day fasting sample). Similarly, Barnes et al. (2016) observed significantly increased excretion of pyrogallol sulfate in human urine after 10 days of MP intake (cv. Keitt, 400 g/day) [20]. Most of the compounds quantified in urine were detected in the plasma. However, we were not able to quantify them in the plasma due to their low concentrations (<LOQ).

To get a better understanding of the changes in the polyphenolic metabolite pools after 14 days consumption of MP, the (poly)phenolic compounds quantified in urine samples were divided into 10 classes: benzoic acid derivatives, phenylacetic acid derivatives, phenylpropanoic acid derivatives, benzaldehyde derivatives, pyrogallol derivatives, catechol derivatives, hippuric acid derivatives, cinnamic acid derivatives, valerolactone derivatives and others (Appendix A, Table A2). The concentrations of polyphenol classes in urine were compared at baseline (0 h, before MP beverage intake), 24 h fasting (24 h after single-time MP beverage intake) and 15th day fasting (24 h after 14 days of MP beverage intake) (Figure 3, Appendix A, Table A2). The concentration increase of pyrogallol derivatives was highest in the 15th day fasting urine samples (3.6% increase) compared to 24 h fasting urine (*p* < 0.05). Minor increases (<1.0%) were also observed in other classes of compounds (Figure 3).

These results are in agreement with previous studies on mango conducted by the Talcott group where they observed significant (*p* < 0.05) increases of pyrogallol and benzoic acid derivatives in urine including methylpyrogallol sulfate, pyrogallol sulfate and methylgallic acid sulfate after consumption of MP [20,52,53]. The increased excretion of mango (poly)phenol (mainly gallotannins) metabolites could be due to the adaptive increase in microbial metabolism after 14 days of daily intake of mango polyphenols [4,33,54]. Generated metabolites support the growth of specific bacterial species, which in turn enhance the production of certain metabolites suggesting reciprocal interaction between them. Several gut microbial species have been associated with gallotannins catabolism. For example, colon microorganisms such as *Lactobacillus plantarum*, *Streptococcus galloylitcus*, *Aspergillus oryzae* and *Lactococcus lactis* can utilize gallotannins and produce gallic acid, pyrogallol and catechol [47,55]. Previous studies in humans demonstrated that repetitive mango supplementation can increase the abundance of pyrogallol-producing microbiota, e.g., *Aspergillus oryzae* and *Lactococcus lactis*, and decrease the abundance of *Bacteroides thetaiotaomicron* and *Clostridium leptum* [47,53]. Additionally, gallotannins and their metabolites may inhibit growth of other bacteria such as *Bacteroides fragilis*, *Escherichia coli*, *Enterobacter cloacae*, *Salmonella typhimurium*, *Salmonella aureus*, etc. [34,35,56], possibly increasing community space for blooming microbiota that can utilize gallotannins and increasing net metabolite concentrations. Enhanced microbial efficiency to metabolize polyphenols may also influence concentration of metabolites [57] although this may not be only true for mango polyphenols, but also for non mango-specific metabolites (i.e., hippuric acid).

### 2.4. Inter-Individual Variability

The inter-individual variability in (poly)phenolic compounds and their metabolites was calculated for baseline (0 h), 24 h fasting and 15th day fasting urine samples after MP beverage intake (Table 3). The percentage of the coefficient of variation (%CV) can show the extent of variability in relation to mean of the population. The highest inter-individual variability in terms of total polyphenols excreted in urine was observed at baseline (172%CV) which decreased tremendously after acute (64%CV) and 14 days (72%CV) of mango intake. The inter-individual variability was also determined in terms of classes of (poly)phenolic metabolites. Benzaldehyde derivatives (217%CV), pyrogallol derivatives (179%CV) and catechol derivatives (182%CV) had the highest %CV in urine at baseline (0 h) which decreased to 57%CV, 81%CV and 73%CV, respectively in 15th day fasting urine samples. On the other hand, the %CV of cinnamic acid derivatives, one of the major phenolic acid groups in mango pulp [58], increased from 68%CV to 136%CV after 14 days of daily intake of mango compared to baseline (Table 3).

Inter-individual variability has been documented previously in metabolite studies and may be attributed to a number of factors including but not limited to the differences in body weight/health status, the composition of individual gut microbiota [59,60], habitual dietary intake as these were free living individuals, host genetics including genetic polymorphism of cytochrome P450 enzymes resulting in an altered expression and function of individual enzymes, and work and living/lifestyle environments, among other influences [61,62].

In this study, we identified and quantified (poly)phenolic compounds in MP, as well as their metabolites after MP consumption in human plasma and urine with advanced instruments such as UHPLC-Q-TOF-MS and UHPLC-QQQ-MS. With our crossover design, we reduced the influence of confounding covariates because each subject served as their own control. However, the study has some limitations. The standards of most metabolites (glucuronides, sulfates etc.) were not commercially available at the time when the study was conducted, so those metabolites were quantified either with the respective parent compounds or with the standards that share similar structures or molecular weight with them. This may lead to differences between their actual concentrations and the reported values.

In summary, this study provides comprehensive characterization of (poly)phenolic metabolites generated after mango consumption in humans both in acute and repetitive intake settings. In addition, this is the first study to explore the effectiveness of the addition of VC for enhancing the absorption/metabolism of MP polyphenols. Understanding the metabolic fate and PK parameters of polyphenols from mangoes will aid in developing future research for assessing potential health benefits associated with mango consumption.

## 3. Materials and Methods

### 3.1. Ethics, Study Design and Study Subjects

The research was conducted at the Clinical Nutrition Research Center (CNRC), Illinois Institute of Technology (IIT) (Chicago, IL, USA). The study was approved by the Institutional Review Board (IRB) and the trial was registered at clinicaltrials.gov (registration number: NCT03365739). All subjects signed the IRB approved informed consent form prior to the start of any study-related procedures. This was a two-part human clinical trial including an acute (24 h) and short-term (14 days) evaluation of MP intake. Thirteen healthy subjects were enrolled and completed both parts of the study protocol (Table 4, Figure 4). Subjects were nonsmokers and were not taking any supplements or medications (i.e., laxatives, proton pump inhibitors etc.) that would interfere with study outcomes. Subjects did not have documented atherosclerotic disease, digestive disorders, inflammatory disease, diabetes mellitus, or other systemic diseases. They did not have any allergy or intolerance to test beverages and study foods. Females of reproductive age were monitored, avoiding the menstruation phase of their menstrual cycle for study day visits.

#### 3.1.1. Acute 24 h Trial

The acute trial was a randomized, 3 arm, within-subject crossover study design (Figure 5A). The trial initiated with 3 days of food record collection to assess background (pre-study) dietary intake followed by counseling to adhere to a diet devoid of mango or its parts and relatively low in polyphenol-rich beverages/foods, which was maintained throughout the duration of the trial. Subjects were also provided a list of “Foods to Avoid” and alternate options to avoid high polyphenol foods and beverages, particularly for the 24 h before each study day, such as avoidance of coffee, tea, caffeinated products and alcohol. In general, subjects were encouraged to follow their “usual” diet with the exception of foods/beverages that may interfere with the outcomes of the study. Food intake diaries and, phone call and email reminders helped subjects maintain compliance throughout the study period. After an initial 7-day run-in period on the limited polyphenol diet, subjects were assigned to a randomization sequence providing 1 of 3 study test beverages (recipes provided in Appendix A, Table A3) on three separate days in a random order: MP (500 g), MP (500 g) + VC (100 mg) or VC (100 mg) beverages. On each study day subjects arrived fasted and well hydrated. After standard admission procedures were completed (e.g., review of compliance with dietary requirements for 3 days prior to study day, dinner meal intake, usual sleep patterns), a catheter was placed by a registered nurse in subjects’ non-dominant arm and a fasting/baseline blood sample was collected (0 h). Subjects were provided with a standard breakfast meal 2 h after the study beverages. The breakfast meal was comprised of a buttermilk biscuit with butter and jelly, and scrambled egg white with shredded white cheddar cheese. After the 6 h blood collection, subjects ate a low polyphenol lunch consisting of dry roasted peanuts and fresh peeled cucumber with ranch dressing (Appendix A, Table A3). Each visit lasted ~10.5 h and subjects were required to remain at the CNRC on the IIT Campus the entire time. After catheters were removed, subjects were given a controlled low polyphenolic dinner meal (provided at every study visit) to eat at home, with reminder instructions for overnight fasting and their scheduled time to report back to the CNRC the next morning for the 24 h blood and urine collection. Blood samples were collected at 0, 0.5, 2, 4, 6, 8, 10 and 24 h and urine samples were collected at 0 and 24 h (Figure 5A).

Blood samples were collected in vacutainers containing ethylenediaminetetraacetic acid (EDTA) and were centrifuged at 425 × *g* for 15 min at 4 °C to separate plasma. Spot urine samples were collected in urine collection cups, immediately placed on ice and then aliquoted into individual cryovials. All the samples were stored at −80 °C until analysis.

#### 3.1.2. Short-Term 14-Day Trial

A 14-day feeding trial was conducted with the MP only to study the effect of repetitive intake of MP on (poly)phenolic metabolite profiles. Upon completion of the acute trial, subjects were instructed to continue on the low polyphenolic diet for 2 weeks while consuming MP beverages (provided by CNRC) daily in the morning (250 g MP + 50 g water) and evening (250 g MP + 50 g water). On day 14, subjects consumed both beverages (500 g of MP in total) in the morning and reported to the CNRC on the 15th day after an overnight fast to provide a fasting blood and urine sample (Figure 5B). Blood and urine samples were collected as described above and stored at −80 °C until analysis.

### 3.2. Chemicals and Materials

HPLC-grade acetonitrile, methanol, acetone, formic acid, acetic acid and polypropylene syringe filters (Whatman^TM^ 0.2 µm) were purchased from Fischer Scientific Co. (Pittsburg, PA, USA). Solid Phase Extraction (SPE) cartridges (C18, 3 mL, 200 mg) were purchased from Agilent Technologies (Santa Clara, CA, USA). Standards of catechol (benzene-1,2-diol), hippuric acid, 2-hydroxyphenylacetic acid, 3-hydroxyphenylpropanoic acid, vanillic acid (4-hydroxy-3-methoxybenzoic acid), 3,4-dihydroxyphenylacetic acid, gallic acid (3,4,5-trihydroxybenzoic acid), ferulic acid (4’-hydroxy-3’-methoxycinnamic acid), 4-hydroxybenzoic acid, caffeic acid (3’,4’-dihydroxycinnamic acid), 2-methyhippuric acid, 4-methyhippuric acid, chlorogenic acid, 4-hydroxybenzaldehyde, 3,4-dihydroxybenzoic acid, 2,3-dihydroxybenzoic acid, 2,5-dihydroxybenzoic acid, *p*-coumaric acid (4’-hydroxycinnamic acid), *m*-coumaric acid (3’-hydroxycinnamic acid), *o*-coumaric acid (2’-hydroxycinnamic acid), catechin and mangiferin were purchased from Fischer Scientific Co. (Pittsburg, PA, USA). The metabolites (glucuronides, sulfates etc.) for which standards are not commercially available were quantified using the standards that share similar structures or molecular weight (the standards used to quantify each compound and their limit of quantification (LOQ) are listed in Appendix A, Table A4). Human blank plasma was purchased from BioIVT (San Francisco, CA, USA). Individually quick frozen (IQF) diced mangoes (variety: Kent) were purchased from Val-Mex Foods (San Antonio, TX, USA) and vitamin C powder (GMO Free Vitamins LLC) was purchased from Amazon.com.

### 3.3. Sample Preparation

#### 3.3.1. Extraction of (Poly)phenolic Compounds from MP

MP (5 g) was extracted with 20 mL of extraction solution (acetone: water: acetic acid = 70:29.7:0.3 *v*/*v*) followed by two subsequent extractions with 10 mL extraction solution. The samples were vortexed for 30 s, followed by 10 mins sonication in iced water. The samples were kept in the dark for 10 mins before centrifugation at 8228 g for 10 min at 4 °C. The supernatants were pooled together after three extractions and final volume was made up to 45 mL. Extract (1 mL) was dried under nitrogen gas and reconstituted in 2 mL of acidified water (0.1% formic acid) for SPE procedure. The elution of the compounds was done with 1 mL of acidified methanol (0.1% formic acid). The eluent was collected and dried under nitrogen gas at room temperature. The dried samples were reconstituted in 200 µL of starting mobile phase (SMP) (0.1% formic acid, 5% acetonitrile in water) and centrifuged at 18,514 × *g* for 10 min at 4 °C. Supernatant was collected in amber HPLC vials before analysis.

#### 3.3.2. Extraction of (Poly)phenolic Compounds from Plasma

SPE procedure was conducted to extract and concentrate the (poly)phenolic compounds and their metabolites from the plasma samples. Briefly, plasma samples were thawed on ice and 500 µL plasma was diluted with 1.5 mL of 0.1% formic acid in water with addition of internal standards (2-methylhippuric acid (2 ng/mL), phloridizin (1 ng/mL)) before being loaded on preconditioned SPE cartridges. The cartridges were washed with 0.1% formic acid in water (1 mL) after loading with plasma samples. The elution of the compounds was achieved by 0.1% formic acid in methanol (1 mL) and the eluent was dried under nitrogen gas at room temperature. The dried samples were reconstituted in 100 µL of SMP and centrifuged at 18,514× *g* for 10 mins at 4 °C. Supernatant was collected in amber HPLC vials before analysis.

#### 3.3.3. Extraction of (Poly)phenolic Compounds from Urine

Urine (500 µL) was diluted and centrifuged at 18,514 × *g* for 10 mins at 4 °C. The supernatant was collected and filtered through 0.2 µm polypropylene syringe filter before HPLC analysis.

### 3.4. Identification and Quantification of (Poly)phenolic Compounds

The (poly)phenolic compounds were identified using an Agilent 1290 Infinity ultrahigh-performance liquid chromatography (UHPLC) system coupled with Agilent 6550 electrospray ionization (ESI) quadrupole time of flight (Q-TOF) mass spectrometer (MS) (Agilent Technologies, Santa Clara, CA, USA). The system was equipped with a binary pump with an integrated vacuum degasser, autosampler with a thermostat and column compartment with a thermostat. Spectra were recorded in negative mode with the following parameters: gas temperature 250 °C, gas flow 10 L/min, nebulizer pressure 35 psi, sheath gas temperature 300 °C, sheath gas flow 11 L/min and capillary 3500 V. MS scan method and Auto MS preferred list of the compounds was created based on personal compound database library (PCDL) and literature reports. Spectra were acquired in MS scan with the *m*/*z* range of 100–1200 and an acquisition rate of 2 spectra per s, and in MS/MS mode with the *m/z* range of 50–1200 and an acquisition rate of 4 spectra per s. The compound identification was based on the MS/MS fragmentation pattern, exact mass (PCDL), retention time match with available standards and previous literature reports. Compounds with mass error less than 5 ppm and/or retention time match with the standards were considered for identification. The data were analyzed using the Mass Hunter Qualitative Analysis software (version B.06.00, Agilent Technologies, Santa Clara, CA, USA). Pooled plasma (*n* = 13) samples were analyzed at 0, 2, 8 and 24 h for identification of (poly)phenolic metabolites.

A UHPLC system coupled with a 6460 Series Triple Quadrupole (QQQ) (Agilent Technologies, Santa Clara, CA, USA) was used for quantitative analysis. The ESI conditions were the same as those used in the UHPLC-Q-TOF analysis. The quantification of the compounds was conducted by multiple-reaction monitoring (MRM) transitions. Spectra were recorded in both positive- and negative-ion mode with capillary voltage of 4500 V and drying gas flow rate of 9 L/min at 200 °C. The sheath gas temperature and flow rate were 300 °C and 11 L/min, respectively. Standards were optimized for collision energies, fragmentor voltages and MRM transitions using Mass Hunter Optimizer. The MRM transition for metabolites were created based on the results from UHPLC-Q-TOF-MS. Standards for plasma analysis were prepared in blank plasma (charcoal-stripped human plasma obtained from Bioreclamation IVT) for matrix match and for urine analysis were prepared in SMP.

(Poly)phenolic compounds and their metabolites in plasma and urine were separated on a Pursuit 3 PFP column (150 × 2.0 mm, 3.0 μm, Agilent Technologies) equipped with a Pursuit MetaGuard column (10 × 2.0 mm, 3.0 μm, Agilent Technologies) at a constant temperature of 35 °C using our previously established method [29]. The flow rate was maintained at 0.4 mL/min and the injection volume was 5 μL. The mobile phase consisted of acidified water (0.1% formic acid) (A) and acidified acetonitrile (0.1% formic acid) (B). The gradient for the separation of compounds was as follows: 5 to 5% B from 0 to 1 min; 10% B at 3 min; 15% B at 7 min; 15% B at 9 min; 20% B at 10 min; 20% B at 11 min; 25% B at 12 min; 30% B at 13 min; 30% B at 14 min; 95% B at 15 min; and back to 5% B at 16 min. The column was re-equilibrated to the initial mobile phase conditions for 4 min before the next injection. The data were analyzed using the Mass Hunter Quantitative Analysis software (version B.07.00, Agilent Technologies, Santa Clara, CA, USA).

### 3.5. Pharmacokinetic and Statistical Analysis

The absorption and elimination of 11 major mango metabolites over 24 h was determined after MP, MP + VC, and VC intake. The plasma concentration/area at each time point was determined by the mean of 13 subjects. The time vs. concentration curves and the time vs. area curves were created using GraphPad Prism (version 8.2.1). C_max_ (nmol/L) is defined as the maximum concentration of (poly)phenolic metabolites in plasma between 0−24 h after test beverage intake. T_max_ is the time (h) when C_max_ was achieved. Wilcoxon signed-rank test was used for T_max_ analysis due to non-normal distribution. Area under the plasma concentration time curve (AUC_0−24h_, nmol·h/L) was calculated by the linear trapezoidal method using Microsoft Excel 2013, version 15 [63].

Subject characteristics were analyzed from data collected at the screening visit and tabulated using descriptive statistics. Results are presented as numbers and percentages, as appropriate. For continuous variables, normality was assessed using Shapiro–Wilk tests, skewness and kurtosis. Data not conforming to normal distribution patterns were log transformed prior to analysis and noted accordingly. Mixed-model repeated measure analysis of covariance (ANCOVA) was performed on each quantitative outcome variable to test main effects of test beverages and time (hour) using PROC MIXED via Window PC-SAS (version 9.4; SAS Institute Inc, Cary, NC). Multiple comparisons within and among test beverages (MP, MP + VC and VC) over 24 h were performed by mixed model statistical significance (*p* < 0.05). A two-tailed distribution, paired t-test (Microsoft Excel 2013, version 15) was conducted for a short-term 14-day feeding trial to compare the (poly)phenolic metabolite concentrations in fasting urine collected after acute (24 h) and 14-day MP intake (15th day). The pharmacokinetic (PK) curves of the (poly)phenolic metabolites with plasma concentrations below LOQ are presented as area vs. time instead of area vs. concentration curves.

The results of the statistical analysis are presented as mean ± standard error unless indicated otherwise. Statistical significance was based on 2-sided test beverage comparison at the 5% significance level under a null hypothesis of no difference between test beverages.

## Figures and Tables

**Figure 1 molecules-25-05522-f001:**
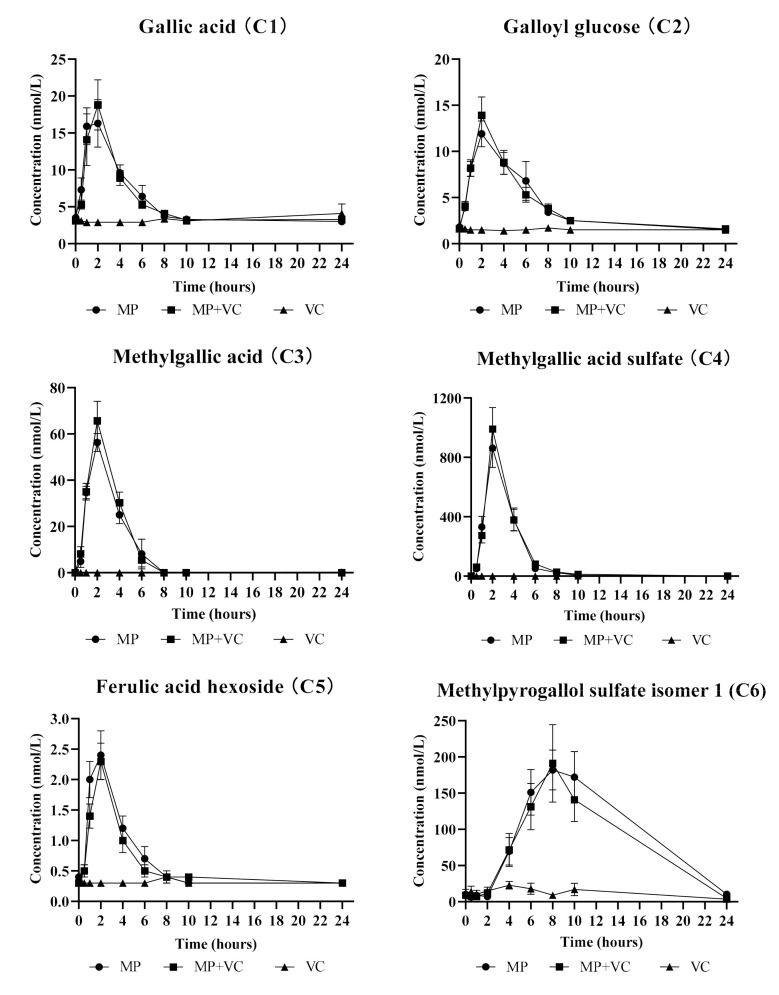
Plasma concentration vs. time profiles (0−24 h) of (poly)phenolic compounds and their metabolites: gallic acid (**C1**), galloyl glucose (**C2**), methylgallic acid (**C3**), methylgallic acid sulfate (**C4**), ferulic acid hexoside (**C5**), methylpyrogallol sulfate isomer 1 (**C6**), methylpyrogallol sulfate isomer 2 (**C7**), pyrogallol sulfate (**C8**), methylpyrogallol glucuronide (**C9**) and catechol glucuronide (**C10**), after test beverages intake. MP (mango pulp), MP + VC (mango pulp + vitamin C) and VC (vitamin C).

**Figure 2 molecules-25-05522-f002:**
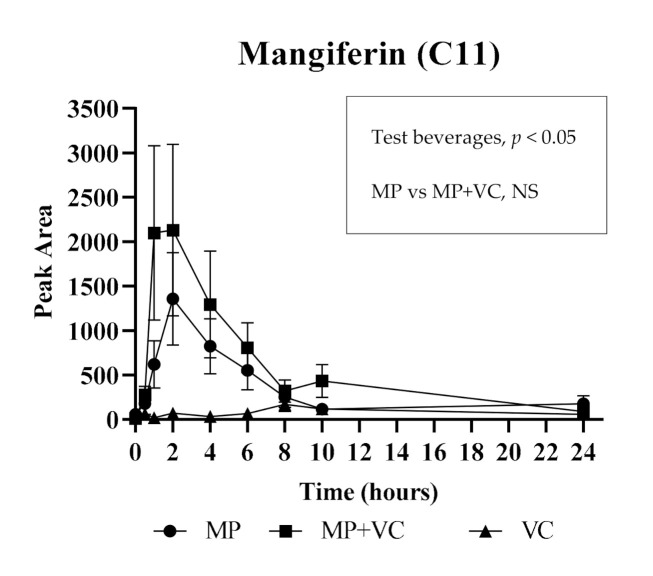
Area vs. time profile (0−24 h) of mangiferin (**C11**) after test beverages intake. The concentration of mangiferin was below the limit of quantitation (LOQ). MP (mango pulp), MP + VC (mango pulp + vitamin C) and VC (vitamin C). NS: not significant.

**Figure 3 molecules-25-05522-f003:**
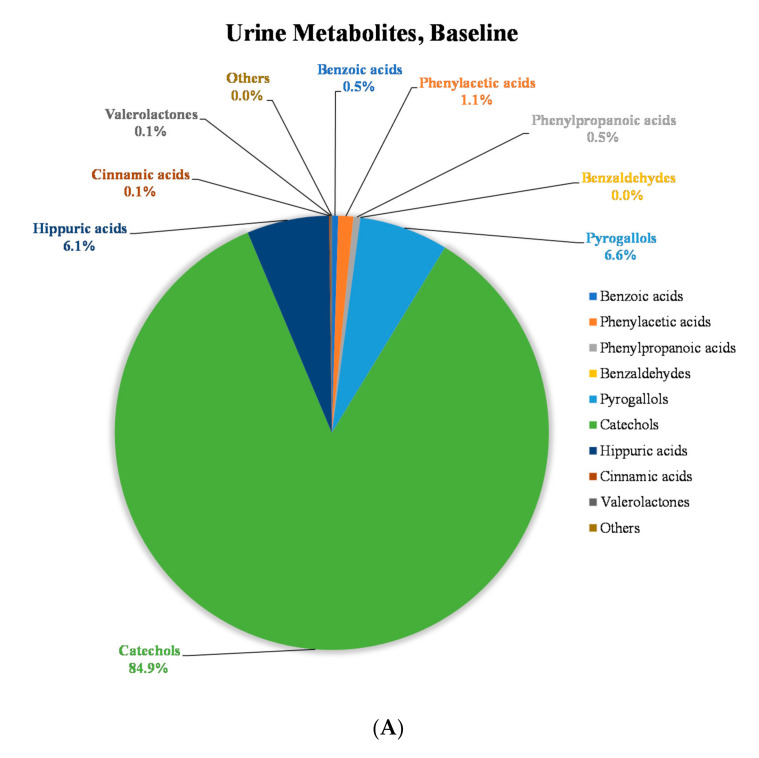
Distribution of concentration of (poly)phenolic metabolites classes in fasting urine at (**A**) baseline (0 h, before MP beverage intake), (**B**) 24 h fasting (24 h after single-time MP beverage intake) and (**C**) 15th day fasting (24 h after 14 days of MP beverage intake).

**Figure 4 molecules-25-05522-f004:**
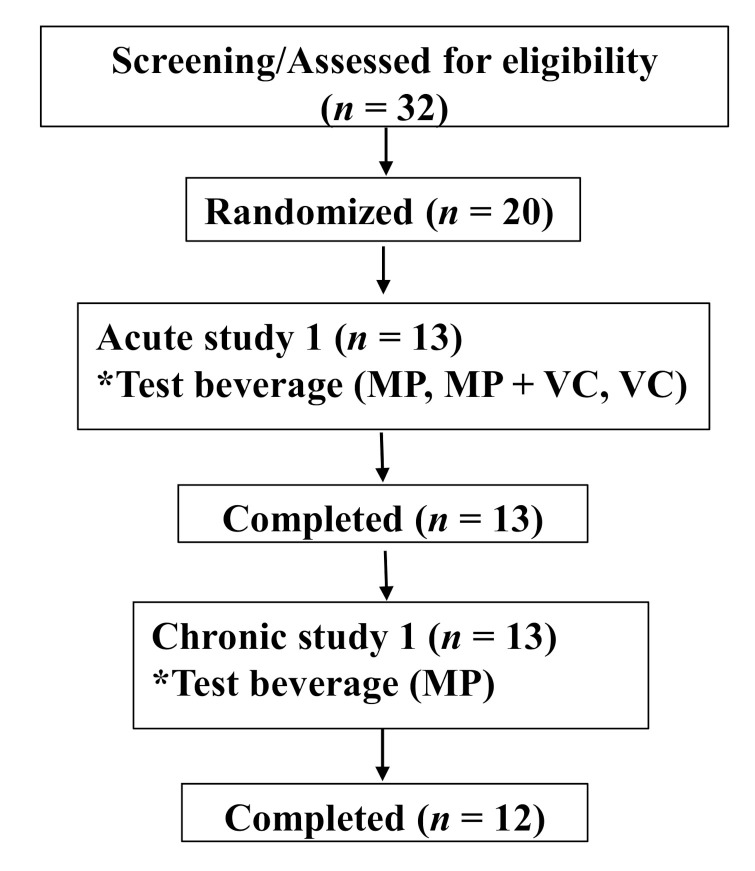
Consolidated Standards of Reporting Trials (CONSORT) flow diagram. * Test beverage: MP (Mango pulp (500 g)), MP+VC (Mango pulp (500 g) + Vitamin C (100 mg)) and VC (Vitamin C (100 mg)).

**Figure 5 molecules-25-05522-f005:**
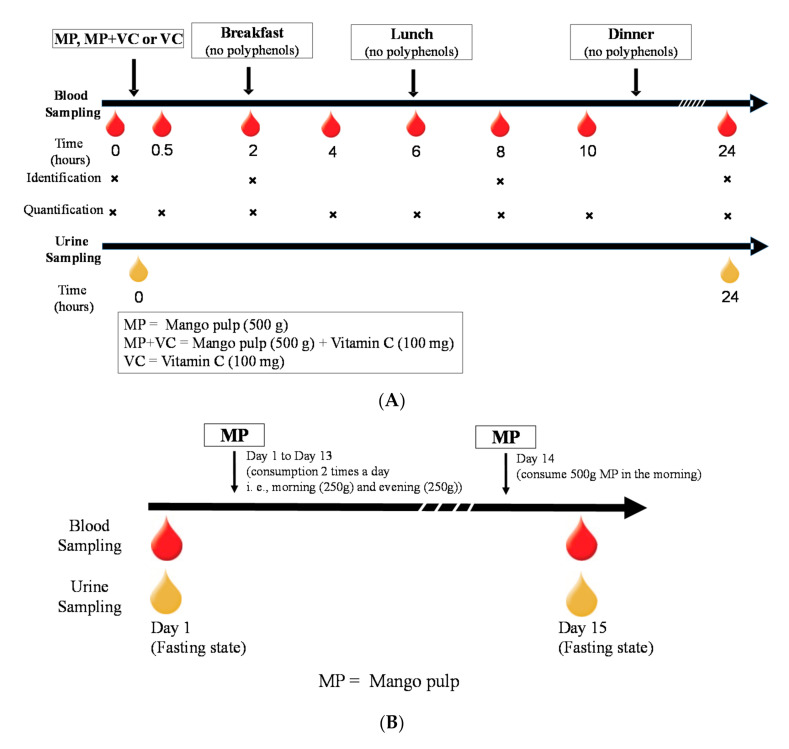
Blood and urine collection scheme on (**A**) an acute study day visit, (**B**) 14-day feeding trial.

**Table 1 molecules-25-05522-t001:** (Poly)phenolic metabolites tentatively identified in plasma at different time points after mango pulp beverage intake (acute trial).

	Compound Number	Retention Time (min)	Observed Ion (*m*/*z*)	Calculated (*m*/*z*)	Mass Error	Molecular Formula	Tentative Compound Identification
2 h	1	1.2	367.10464	367.10287	4.8	C_17_H_20_O_9_	Feruloylquinic acid
	2	4.5	183.02976	183.02937	2.1	C_8_H_8_O_5_	Methylgallic acid
	3	4.7	167.03452	167.03447	0.3	C_8_H_8_O_4_	Vanillic acid
	4	6.5	285.06163	285.06107	2.0	C_12_H_14_O_8_	Catechol glucuronide
	5	9.2	299.07736	299.07667	2.3	C_13_H_16_O_8_	Hydroxybenzoic acid glucoside
	6	14.2	471.18600	471.18667	−1.4	C_22_H_32_O_11_	Abscisic acid glucose ester (formate adduct)
	7	14.4	192.06678	192.06607	3.7	C_10_H_11_NO_3_	Methylhippuric acid
8 h	1	4.5	315.07213	315.07157	1.8	C_13_H_16_O_9_	Methylpyrogallol glucuronide
	2	5.3	371.09754	371.09787	−0.9	C_16_H_20_O_10_	Methoxyphenylpropanoic acid glucuronide
	3	6.0	204.98140	204.98067	3.6	C_6_H_6_O_6_S	Pyrogallol sulfate
	4	8.9	218.99631	218.99637	−0.3	C_7_H_8_O_6_S	Methylpyrogallol sulfate
	5	9.3	299.07739	299.07667	2.4	C_13_H_16_O_8_	Hydroxybenzoic acid glucoside
	6	22.1	331.0668	331.06657	3.7	C_13_H_16_O_10_	Galloyl glucose
24 h	1	5.3	194.04533	194.04537	−0.2	C_9_H_9_NO_4_	Hydroxyhippuric acid
	2	6.4	341.08881	341.08727	4.5	C_15_H_18_O_9_	Hydroxyphenylproanoic acid glucuronide
	3	7.9	218.99631	218.99637	−0.3	C_7_H_8_O_6_S	Methylpyrogallol sulfate
	4	15.9	194.04602	194.04537	3.3	C_9_H_9_NO_4_	Hydroxyhippuric acid isomer
	5	20.7	597.10752	597.10917	−2.8	C_25_H_26_O_17_	Iso/mangiferin glucuronide

These metabolites were not observed at time point *t* = 0 h (baseline i.e., before mango intake).

**Table 2 molecules-25-05522-t002:** Pharmacokinetic parameters (0−24 h) and *p* values of (poly)phenolic compounds and their metabolites in human plasma after ingestion of MP, MP + VC and VC test beverages (mean ± standard error).

Compound Names (Number)	Pharmacokinetic Parameters	MP (A)	MP + VC (B)	VC (C)	*p* Value			
					Tx	A vs. B	A vs. C	B vs. C
Gallic acid (**C1)**	C_max_ (nmol/L)	20.4 ± 3.1	21.1 ± 3.9	5.3 ± 1.3	<0.0001	NS	<0.0001	<0.0001
	T_max_ (h)	2.2 ± 0.4	2.0 ± 0.3	N/A		NS	N/A	N/A
	AUC_0–24h_ (nmol·h/L)	64.5 ± 7.8	62.8 ± 6.9	24.9 ± 1.9	<0.0001	NS	<0.0001	<0.0001
Galloyl glucose (**C2**)	C_max_ (nmol/L)	13.4 ± 1.9	14.0 ± 2.0	2.1 ± 0.3	<0.0001	NS	<0.0001	<0.0001
	T_max_ (h)	2.4 ± 0.3	2.2 ± 0.2	N/A		NS	N/A	N/A
	AUC_0–24h_ (nmol·h/L)	46.1 ± 5.0	48.0 ± 5.6	12.3 ± 0.7	<0.0001	NS	<0.0001	<0.0001
Methylgallic acid (**C3**)	C_max_ (nmol/L)	54.5 ± 4.9	65.7 ± 8.4	4.8 ± 0.7	<0.0001	NS	<0.0001	<0.0001
	T_max_ (h)	2.2 ± 0.3	2.0 ± 0.0	N/A		NS	N/A	N/A
	AUC_0–24h_ (nmol·h/L)	161.1 ± 13.6	178.7 ± 17.5	25.9 ± 1.9	<0.0001	NS	<0.0001	<0.0001
Methylgallic acid sulfate (**C4**)	C_max_ (nmol/L)	824.2 ± 130.8	990.8 ± 145.0	6.3 ± 1.6	<0.0001	NS	<0.0001	<0.0001
	T_max_ (h)	2.2 ± 0.2	2.0 ± 0.0	N/A		NS	N/A	N/A
	AUC_0–24h_ (nmol·h/L)	1655.8 ± 274.0	1837.1 ± 258.3	24.7 ± 3.0	<0.0001	NS	<0.0001	<0.0001
Ferulic acid hexoside (**C5**)	C_max_ (nmol/L)	2.3 ± 0.4	2.4 ± 0.3	0.5 ± 0.1	<0.0001	NS	<0.0001	<0.0001
	T_max_ (h)	1.8 ± 0.2	1.8 ± 0.1	N/A		NS	N/A	N/A
	AUC_0–24h_ (nmol·h/L)	7.7 ± 1.2	6.8 ± 0.7	2.5 ± 0.1	<0.0001	NS	<0.0001	<0.0001
Methylpyrogallol sulfate isomer (**C6**)	C_max_ (nmol/L)	230.2 ± 32.3	224.9 ± 50.3	35.8 ± 9.5	<0.0001	NS	<0.0001	<0.0001
	T_max_ (h)	8.6 ± 0.5	7.7 ± 0.6	N/A		NS	N/A	N/A
	AUC_0–24h_ (nmol·h/L)	616.9 ± 83.7	578.1 ± 103.7	129.5 ± 36.0	<0.0001	NS	<0.0001	<0.0001
Methylpyrogallol sulfate isomer (**C7**)	C_max_ (nmol/L)	243.5 ± 71.8	232.8 ± 69.8	76.6 ± 24.9	<0.05	NS	<0.05	<0.05
	T_max_ (h)	9.8 ± 1.8	6.3 ± 1.0	N/A		NS	N/A	N/A
	AUC_0–24h_ (nmol·h/L)	375.4 ± 94.3	394.6 ± 89.1	216.6 ± 54.5	<0.05	NS	<0.05	<0.05
Pyrogallol sulfate (**C8**)	C_max_ (nmol/L)	10023.8 ± 1781.1	8823.4 ± 1767.4	2564.2 ± 893.5	<0.05	NS	<0.05	<0.05
	T_max_ (h)	7.3 ± 0.6	7.3 ± 0.7	N/A		NS	N/A	N/A
	AUC_0–24h_ (nmol·h/L)	18852.1 ± 3491.6	17956.9 ± 3093.6	5504.9 ± 1763.4	<0.05	NS	<0.05	<0.05
Methylpyrogallol glucuronide (**C9**)	C_max_ (nmol/L)	6.7 ± 2.3	4.5 ± 0.3	4.8 ± 0.3	F	F	F	F
	T_max_ (h)	7.2 ± 2.4	9.7 ± 2.4	8.0 ± 2.3		NS	N/A	N/A
	AUC_0–24h_ (nmol·h/L)	29.1 ± 3.0	26.7 ± 1.8	26.1 ± 1.7	NS	NS	NS	NS
Catechol glucuronide (**C10**)	C_max_ (nmol/L)	471.7 ± 270.2	185.8 ± 45.8	10.6 ± 3.2	<0.0001	NS	<0.0001	<0.0001
	T_max_ (h)	8.0 ± 0.6	8.3 ± 0.4	N/A		NS	N/A	N/A
	AUC_0–24h_ (nmol·h/L)	1830.9 ± 1391.0	483.3 ± 108.0	44.8 ± 5.1	<0.0001	NS	<0.0001	<0.0001

Test beverages—A: MP (mango pulp), B: MP + VC (mango pulp + vitamin C), C: VC (vitamin C). Tx: test beverage effect, NS: not significant, N/A: not applicable, F: failed normality test.

**Table 3 molecules-25-05522-t003:** Coefficient of variation (%CV) of (poly)phenolic metabolites in fasting urine samples after acute and 14 days of mango beverage consumption.

	0 h	24 h	Day 15
Benzoic acid derivatives	47	49	65
Phenylacetic acid derivatives	59	51	71
Phenylpropanoic acid derivatives	48	57	76
Benzaldehyde derivatives	217	98	57
Pyrogallol derivatives	179	74	81
Catechol derivatives	182	64	73
Hippuric acid derivatives	61	50	36
Cinnamic acid derivatives	68	38	136
Valerolactone derivatives	112	117	112
Others	85	47	51
Total polyphenols	172	64	72

Coefficient of variation (%CV) (standard deviation/mean concentration × 100%).

**Table 4 molecules-25-05522-t004:** Subject demographic characteristics.

Variable	Acute Trial (*n* = 13)	14-Day Feeding Trial (*n* = 12)
Age (year)	30 ± 2	31 ± 2
Height (cm)	169.5 ± 2.1	169.4 ± 2.2
Weight (kg)	64.9 ± 1.9	65.0 ± 2.0
BMI (kgm^−2^)	22.7 ± 0.4	22.7 ± 0.5
Heart rate (beats per minute)	68 ± 2	69 ± 2
Systolic blood pressure (mmHg)	115 ± 2	115 ± 2
Diastolic blood pressure (mmHg)	73 ± 2	74 ± 2
Mid-point waist circumference (cm)	78.8 ± 1.1	78.8 ± 1.2
Male:Female	7:6	6:6

Values are presented as mean ± standard error.

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
