# Peer review of "Pharmacokinetic Characterization of (Poly)phenolic Metabolites in Human Plasma and Urine after Acute and Short-Term Daily Consumption of Mango Pulp"

_molecules, 2020, doi:10.3390/molecules25235522_

Round 1

Reviewer 1 Report

The manuscript titled “Pharmacokinetic Characterization of Polyphenolic Metabolites in Human Plasma and Urine after Acute and Short-term Daily Consumption of Mango Pulp” is a very well written manuscript. There is extensive amount of data as supplemental file which is good.

There are few minor comments that need to be addressed:

  1. Authors did not mention about the pharmacokinetic model used in data analyses and there is little or no description of the pharmacokinetic analyses method. Authors need to expand the method part.
  2. There is discussion of interindividual variability but cytochrome P450 enzymes have not been discussed as a potential reason of variability among individuals.

Reviewer 2 Report

In my opinion, this work is solid and can be accepted after minor revision.

Abstract: Please avoid the use of abbreviations in this section

Reviewer 3 Report

There are very elegant study introducing very important data on phenolic metabolites in human plasma and urine after acute and chronic mango pulp consumption. However, a few points should be clarified:

Table 1- It seems that the first 8 metabolites were detected after 2 h, however, it is not indicated

Fig.1 -  It is indicated that differences between AUCs are not significant. However, for C^, C7 and C8 AUCs for VC  seem to be significantly different as compared to AUCs for MP and MP+VC

Fig.4 – Presentation of “Acute 2  study” on this flow diagram is misleading for a reader who is looking for “Acute 2 study “ data.  As Acute 2 study is a subject of another manuscript,  “Acute 2 study” should not be indicated on Fig.4  
